# WeightedSHAP: analyzing and improving Shapley based feature attributions

**Yongchan Kwon**
Columbia University
New York, NY, 10027
yk3012@columbia.edu

**James Zou**
Stanford University and Amazon AWS
Stanford, CA, 94305
jamesz@stanford.edu

## Abstract

Shapley value is a popular approach for measuring the influence of individual features. While Shapley feature attribution is built upon desiderata from game theory, some of its constraints may be less natural in certain machine learning settings, leading to unintuitive model interpretation. In particular, the Shapley value uses the same weight for all marginal contributions—i.e. it gives the same importance when a large number of other features are given versus when a small number of other features are given. This property can be problematic if larger feature sets are more or less informative than smaller feature sets. Our work performs a rigorous analysis of the potential limitations of Shapley feature attribution. We identify simple settings where the Shapley value is mathematically suboptimal by assigning larger attributions for less influential features. Motivated by this observation, we propose WeightedSHAP, which generalizes the Shapley value and learns which marginal contributions to focus directly from data. On several real-world datasets, we demonstrate that the influential features identified by WeightedSHAP are better able to recapitulate the model's predictions compared to the features identified by the Shapley value.

## 1 Introduction

Explaining how a feature impacts a model prediction is a crucial question in machine learning (ML) as it provides a deeper understanding of how the model behaves and what insights have been extracted from data. In many real-world applications, it has been increasingly common to deploy complicated models such as a deep neural network model or a random forest to achieve high predictability. However, it often comes with a cost of unintuitive interpretations, and it naturally calls for a principled and practical attribution method. The goal of this work is to quantify the contribution of individual features to a particular prediction, also known as the attribution problem.

Lundberg and Lee [2017] proposed a model-agnostic attribution method, SHapley Additive exPlanations (SHAP), based on the Shapley value from economics. Supported by theoretical properties that the Shapley value satisfies, SHAP has been a popular method in the attribution literature [Janzing et al., 2020, Sundararajan and Najmi, 2020]. For instance, Frye et al. [2020] and Aas et al. [2021] developed the SHAP algorithms for dependent features, and Heskes et al. [2020] and Wang et al. [2021] proposed a rigorous framework in causal inference settings. One of the practical problems to use SHAP is the heavy computational costs, and there have been works on improving computational efficiency [Covert and Lee, 2021, Lundberg et al., 2020, Jethani et al., 2021a]. In addition, SHAP has been deployed to various applied scientific research [Lundberg et al., 2018, Janizek et al., 2021, Qiu et al., 2022].

The Shapley value, a mathematical basis of SHAP, is a simple average of the marginal contributions that quantify the average change in a coalition function when a feature of interest is added from a

subset of features with a given coalition size. There are different version of the marginal contributions by the coalition size, and the Shapley value takes a uniform weight to summarize the influence of a feature. This uniform weight arises due to the efficiency axiom of the Shapley value, which requires the sum of attributions to equal the original model prediction, but it is often problematic because some may be more informative than others. As we will show in Section 3, the Shapley value is not optimal to sort features in order of influence on a model prediction.

**Our contributions.** While SHAP is widely used for feature attribution, its limitations are still not rigorously understood. We first show the suboptimality of the Shapley value through an analysis of the marginal contributions. We identify a key limitation of the Shapley value in that it assigns uniform weights to marginal contributions in computing the attribution score. We show that this can lead to attribution mistakes when different marginal contributions have different signal and noise. Motivated by this analysis, we propose WeightedSHAP, a generalization of the Shapley value which is more flexible. WeightedSHAP uses a weighted average of marginal contributions where the weights can be learned from the data. On several real-world datasets, our experiments demonstrate that WeightedSHAP is better able to identify influential features that recapitulate a model's prediction compared to a standard SHAP. WeightedSHAP is a simple modification of SHAP and is easy to adapt from existing SHAP implementations.

## 1.1 Related works

**Model interpretation** There are mainly two types of model interpretation depending on the quantity to be accounted for; global and local interpretations. The global interpretation is to explain the impact of a feature on a prediction model across the entire dataset [Lipovetsky and Conklin, 2001, Breiman, 2001, Owen, 2014, Broto et al., 2020, Zhao and Hastie, 2021, Bénard et al., 2022]. For instance, for a decision tree model, Breiman et al. [2017] measures the total decrease of node impurity at node split by a feature of interest as an impact. In contrast, the local interpretation is to explain the impact of a feature on a particular prediction value [Lundberg and Lee, 2017, Chen et al., 2018, Janzing et al., 2020, Lundberg et al., 2020, Heskes et al., 2020, Jethani et al., 2021a]. For a deep neural network model, a gradient-based method uses the gradient evaluated at a particular input sample as an impact [Simonyan et al., 2013, Sundararajan et al., 2017, Ancona et al., 2017, Selvaraju et al., 2017, Adebayo et al., 2018]. Our work studies the local interpretation problem with a focus on a marginal contribution-based method, which we review in Section 2. The marginal contribution-based method is potentially advantageous over a gradient-based method as it does not require the differentiability of a prediction model.

**Shapley value and its extension** The Shapley value, introduced as a fair division method from economics [Shapley, 1953a], has been deployed in various ML problems. One leading application is data valuation, where the main goal is to quantify the impact of individual data points in model training. Ghorbani and Zou [2019] and Jia et al. [2019] propose to use the Shapley value for measuring the individual data value, and this concept has been extended to handle the randomness of data [Ghorbani et al., 2020, Kwon et al., 2021]. As for the other applications of the Shapley value, model explainability [Ghorbani and Zou, 2020], model valuation [Rozemberczki and Sarkar, 2021], federated learning [Liu et al., 2021], and multi-agent reinforcement learning [Li et al., 2021] have been studied. We refer to Rozemberczki et al. [2022] for a complementary literature review of ML applications of the Shapley value.

The relaxation of the Shapley axioms has been one of the central topics in cooperative game theory [Shapley, 1953b, Banzhaf III, 1964, Kalai and Samet, 1987, Weber, 1988]. Recently, Kwon and Zou [2021] propose to relax the efficiency axiom in the data valuation problem, showing promising results in the low-quality data detection task. Given that Shapley axioms are often not readily applicable to ML problems, relaxing them has the potential to capture a better notion of significance. In this work, we explore the benefits of relaxation of the efficiency axiom on the attribution problem.

## 2 Preliminaries

We review the marginal contribution and the Shapley value in the context of an attribution problem. We first introduce some notations. For $d \in \mathbb{N}$, let $\mathcal{X} \subseteq \mathbb{R}^d$ and $\mathcal{Y} \subseteq \mathbb{R}$ be an input space and an output space, respectively. We use a capital letter $X = (X_1, \ldots, X_d)$ for an input random

variable defined on $\mathcal{X}$, and a lower case letter $x = (x_1, \ldots, x_d)$ for its realized value. We denote a prediction model by $\hat{f} : \mathcal{X} \to \mathcal{Y}$. For $j \in \mathbb{N}$, we set $[j] := \{1, \ldots, j\}$ and denote a power set of $[j]$ by $2^{[j]}$. For a vector $u \in \mathbb{R}^d$ and a subset $S = (j_1, \ldots, j_{|S|}) \subseteq [d]$, we denote a subvector by $u_S := (u_{j_1}, \ldots, u_{j_{|S|}})$. We assume that $X$ has a joint distribution $p(X)$ such that a conditional distribution $p(X_{[d]\backslash S} \mid X_S)$ is well-defined for any subset $S \subsetneq [d]$. With the notations, a conditional coalition function $v_{x,\hat{f}}^{(\text{cond})} : 2^{[d]} \to \mathbb{R}$ is defined as follows [Lundberg and Lee, 2017].

$$v_{x,\hat{f}}^{(\text{cond})}(S) := \mathbb{E}[\hat{f}(x_S, X_{[d]\backslash S}) \mid X_S = x_S] - \mathbb{E}[\hat{f}(X)], \tag{1}$$

where the first expectation is taken with a conditional distribution $p(X_{[d]\backslash S} \mid X_S = x_S)$ and the second expectation is taken with a joint distribution $p(X)$. Here, we use a slight abuse of notation for $\hat{f}(x_S, X_{[d]\backslash S})$ to describe $f(u)$ where $u_i = x_i$ if $i \in S$, and $u_i = X_i$ otherwise. By convention, we set $v_{x,\hat{f}}^{(\text{cond})}([d]) := \hat{f}(x) - \mathbb{E}[\hat{f}(X)]$ and $v_{x,\hat{f}}^{(\text{cond})}(\emptyset) := 0$. A conditional coalition function defined in Equation (1) is a prediction recovered after observing partial information $x_S$ compared to the null information. For instance, if $S = \emptyset$, the first term becomes the marginal expectation $\mathbb{E}[\hat{f}(X)]$ and nothing is recovered by $S = \emptyset$. For ease of notation, we write $v_{x,\hat{f}}^{(\text{cond})}(S) = v^{(\text{cond})}(S)$ for remaining part of the paper.

**Marginal contribution-based attribution methods**  Given that the goal of the attribution problem is to assign the significance of an individual feature $x_i$ on the prediction $\hat{f}(x)$, its primary challenge is how to measure the influence of the feature $x_i$. A leading approach is to quantify the difference in the conditional coalition function values $v^{(\text{cond})}$ after adding one feature of interest. We formalize this concept below.

**Definition 1** (Marginal contribution). *For $i, j \in [d]$, we define the marginal contribution of the $i$-th feature $x_i$ with respect to $j - 1$ features as follows.*

$$\Delta_j(x_i) := \frac{1}{\binom{d-1}{j-1}} \sum_{S \subseteq [d]\backslash\{i\}, |S|=j-1} v^{(\text{cond})}(S \cup \{i\}) - v^{(\text{cond})}(S). \tag{2}$$

The marginal contribution $\Delta_j(x_i)$ considers every possible subset $S \subseteq [d]\backslash\{i\}$ with the coalition size $|S| = j - 1$ and takes a simple average of the difference $v^{(\text{cond})}(S \cup \{i\}) - v^{(\text{cond})}(S)$. That is, it measures the average contribution of the $i$-th feature $x_i$ when it is added to a subset $S$.

Different marginal contributions $\Delta_j(x_i)$ have been studied depending on the coalition size $j$ in the literature. Zintgraf et al. [2017] considered $j = d$ and measured the leave-one-out marginal contribution $\Delta_d(x_i) = \hat{f}(x) - \mathbb{E}[\hat{f}(x_{[d]\backslash\{i\}}, X_i) \mid X_{[d]\backslash\{i\}} = x_{[d]\backslash\{i\}}]$ as an influence of a feature. Guyon and Elisseeff [2003] considered $j = 1$ and measured the coefficient of determination as an influence of a feature. Although they did not use an individual prediction, their idea is essentially similar to using $\Delta_1(x_i) = \mathbb{E}[\hat{f}(x_i, X_{[d]\backslash\{i\}}) \mid X_i = x_i] - \mathbb{E}[\hat{f}(X)]$.

Another widely used marginal contribution-based method is the Shapley value [Lundberg and Lee, 2017, Covert and Lee, 2021]. It summarizes the impact of one feature by taking a simple average across all marginal contributions. To be more specific, the Shapley value is defined as follows.

$$\phi_{\text{shap}}(x_i) := \frac{1}{d} \sum_{j=1}^{d} \Delta_j(x_i). \tag{3}$$

The Shapley value in (3) is known as the unique function that satisfies the four axioms of a fair division in cooperative game theory [Shapley, 1953a]. The four axioms and the uniqueness of the Shapley value are discussed in more detail in Appendix.

Although the Shapley value provides a principled framework in game theory, one critical issue is that the economic notion of the Shapley axioms is not intuitively applicable to the attribution problem [Kumar et al., 2020, Rozemberczki et al., 2022]. In particular, the efficiency axiom, which requires the sum of the attributions to be equal to $v^{\text{cond}}([d])$, is not necessarily essential because an order of attributions is invariant to the constant multiplication. For instance, for any positive constant $C > 0$,

an attribution $\phi_C(x_i) := C \times \phi_{\text{shap}}(x_i)$ will have the same order as the Shapley value $\phi_{\text{shap}}$, but the efficiency axiom is not required for $\phi_C$. In Section 4, we will revisit this point and introduce a new attribution method that relaxes the efficiency axiom.

**Evaluation metrics for the attribution problem.** In the literature, different notions of goodness have been proposed, for instance, the complete axiom [Sundararajan et al., 2017, Shrikumar et al., 2017], the local Lipschitzness [Alvarez-Melis and Jaakkola, 2018], and the explanation infidelity [Yeh et al., 2019] with a focus on the total sum of attributions or the sensitivity of attributions. Recently, Jethani et al. [2021a] suggested using the *Inclusion AUC* to assess the goodness of an order of attributions. Specifically, the Inclusion AUC is measured as follows: Given an attribution method, features are first ranked based on their attribution values. Then the area under the receiver operating characteristic curve (AUC) is iteratively evaluated by adding features one by one from the most influential to the least influential. This procedure generates a AUC curve as a function of the number features added, and the Inclusion AUC is defined as the area under this curve. Similar evaluation metrics have been used in Petsiuk et al. [2018] and Lundberg et al. [2020]. Following the literature, we consider the area under the prediction recovery error curve (AUP) defined as follows.

**Definition 2** (Area under the prediction recovery error curve). *For a given attribution method $\phi$, an input $x \in \mathcal{X}$, and $k \in [d]$, let $\mathcal{I}(k; \phi, x) \subseteq [d]$ be a set of $k$ integers that indicates $k$ most influential features based on their absolute value $|\phi(x_j)|$. For a prediction model $\hat{f}$, we define the area under the prediction recovery error curve at $x$ as follows.*

$$\text{AUP}(\phi; x, \hat{f}) := \sum_{k=1}^{d} \left| \hat{f}(x) - \mathbb{E}[\hat{f}(X) \mid X_{\mathcal{I}(k; \phi, x)} = x_{\mathcal{I}(k; \phi, x)}] \right|. \tag{4}$$

AUP is defined as the sum of the absolute differences between the original prediction $\hat{f}(x)$ and its conditional expectation $\mathbb{E}[\hat{f}(X) \mid X_{\mathcal{I}(k; \phi, x)} = x_{\mathcal{I}(k; \phi, x)}]$ when the $k$ most influential features are given. Each term in Equation (4) measures the amount of a prediction that is not recovered by the $k$ most influential features, and thus this prediction recovery error is expected to decrease as $k$ increases. The prediction recovery error can be described as a function of $k$, and the AUP measures the area under this function as in the Inclusion AUC.

## 3 The Shapley value is suboptimal

In this section, we show that the suboptimality of the Shapley value through a rigorous analysis of the marginal contribution. We first derive a useful closed-form expression of the marginal contribution when $\hat{f}$ is linear and $p(X)$ is Gaussian (Section 3.1). With this theoretical result, we present two simulation experiments where the Shapley value incorrectly reflects the influence of features, resulting in a suboptimal order of attributions (Section 3.2).

### 3.1 A closed-form expression of the marginal contribution

To this end, we assume that a prediction model $\hat{f}$ is linear and an input distribution $p(X)$ follows a Gaussian distribution with zero mean and a block diagonal covariance matrix. To be more specific, we define some notations. For $B \in \mathbb{N}$, we set a vector $\mathbf{d} = (d_1, \ldots, d_B) \in \mathbb{N}^B$ such that $\sum_{b=1}^{B} d_b = d$ and a vector $\boldsymbol{\rho} := (\rho_1, \ldots, \rho_B) \in [0, 1)^B$. We denote a $d \times d$ block diagonal covariance matrix by $\Sigma_{\boldsymbol{\rho}, \mathbf{d}}^{(\text{block})} = \text{diag}\left( \Sigma_{(\rho_1, d_1)}, \ldots, \Sigma_{(\rho_B, d_B)} \right)$ where $\Sigma_{(\rho_b, d_b)} = (1 - \rho_b) I_{d_b} + \rho_b \mathbb{1}_{d_b} \mathbb{1}_{d_b}^T$. Here, for $j \in \mathbb{N}$, we denote the $j \times j$ identity matrix by $I_j$, the $j$-dimensional vector of ones by $\mathbb{1}_j := (1, \ldots, 1)^T \in \mathbb{R}^j$ and $\mathbf{0}_j := 0 \times \mathbb{1}_j$. Lastly, we denote a Gaussian distribution with a mean vector $\mu$ and a covariance matrix $\Sigma$ by $\mathcal{N}(\mu, \Sigma)$. With the notations, we assume $X \sim \mathcal{N}(\mathbf{0}_d, \Sigma_{\boldsymbol{\rho}, \mathbf{d}}^{(\text{block})})$. That is, every feature is normalized to have a unit variance and is included in one of $B$ independent clusters. For $j \in [B]$, the size of the $j$-th cluster is $d_j$, and features are equally correlated to each other within a cluster. The correlation levels can vary from cluster to cluster.

In general, the marginal contribution in Equation (2) does not have a closed-form expression, and it makes a rigorous analysis of the Shapley value difficult. In the following theorem, we derive a closed-form expression of the marginal contribution when $\hat{f}$ is linear and $X \sim \mathcal{N}(\mathbf{0}_d, \Sigma_{\boldsymbol{\rho}, \mathbf{d}}^{(\text{block})})$.

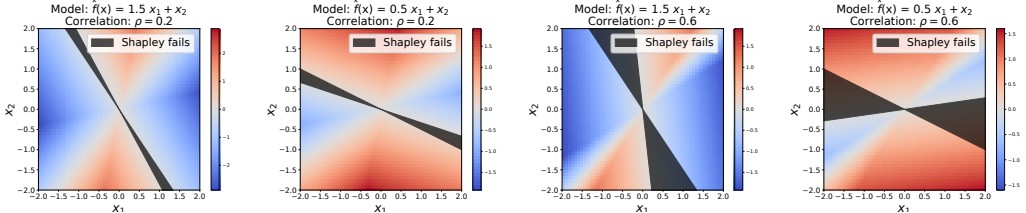

(a) Illustrations of the suboptimality of Shapley-based feature attributions on the four different situations when $d = 2$.

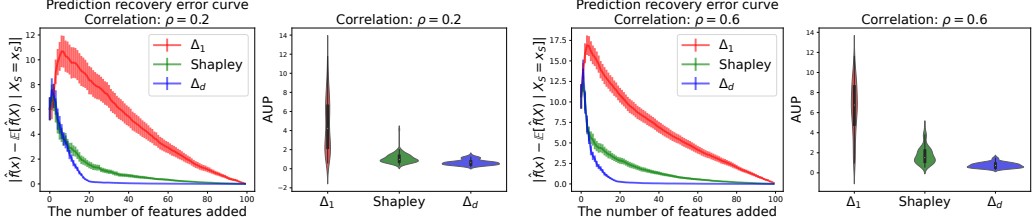

(b) Illustrations of a prediction recovery error curve and AUP comparison when $d = 100$.

Figure 1: The Shapley value is suboptimal. (Top) a region described in black denotes the area the Shapley value fails to select the more influential feature. We encode $\mathcal{E}(1; x, \hat{f}) - \mathcal{E}(2; x, \hat{f})$ as the background color to visualize which feature is more influential. Blue color describes a region where the first feature $x_1$ is more influential, *i.e.*, $\mathcal{E}(1; x, \hat{f}) < \mathcal{E}(2; x, \hat{f})$, and red describes a region where the second feature $x_2$ is more influential, *i.e.*, $\mathcal{E}(1; x, \hat{f}) > \mathcal{E}(2; x, \hat{f})$. The intensity for $\mathcal{E}(1; x, \hat{f}) - \mathcal{E}(2; x, \hat{f})$ is described in a color bar. (Bottom) we compare the three attribution methods $\Delta_1, \phi_{\text{shap}}$, and $\Delta_d$ on the two different situations by varying correlation $\rho \in \{0.2, 0.6\}$. As for the prediction recovery error curve, we denote a 95% confidence band based on 100 samples. The lower AUP is, the better. In both settings, the Shapley value is suboptimal according to AUP.

**Theorem 1** (A closed-form expression for the marginal contribution). *Suppose $\hat{f}(x) = \hat{\beta}_0 + x^T \hat{\beta}$ for some $(\hat{\beta}_0, \hat{\beta}) \in \mathbb{R} \times \mathbb{R}^d$ and $X \sim \mathcal{N}(\mathbf{0}_d, \Sigma_{\boldsymbol{\rho}, \mathbf{d}}^{(\text{block})})$. Then, for $i, j \in [d]$, the marginal contribution of the $i$-th feature $x_i$ with respect to $j - 1$ samples is expressed as*

$$\Delta_j(x_i) = x^T H(i, j)\hat{\beta},$$

*for some explicit matrix $H(i, j) \in \mathbb{R}^{d \times d}$.*

A proof and the explicit term for $H(i, j)$ are provided in Appendix. Theorem 1 shows that the marginal contribution is a bilinear function of an input $x$ and the estimated regression coefficient $\hat{\beta}$. One direct consequence is that the Shapley value also has a bilinear form $\phi_{\text{shap}}(x_i) = x^T H(i)\hat{\beta}$ for $H(i) := \sum_{j=1}^{d} H(i, j)/d$. We emphasize that this bilinear form greatly improves computational efficiency. Specifically, for all $i, j \in [d]$, since the term $H(i, j)\hat{\beta}$ is not a function of an input $x$, we only need to compute the one-time in multiple attribution computations. Moreover, it also leads to a memory efficient algorithm as there is no need to store the $d \times d$ matrix $H(i, j)$.

### 3.2 Motivational examples

With the theoretical result introduced in the previous subsection, we show that the Shapley-based feature attribution is suboptimal and fails to assign larger values to more influential features.

**When there are two features.** When $d = 2$, there are only two possible values for AUP. For any attribution method $\phi$,

$$\text{AUP}(\phi; x, \hat{f}) = \begin{cases} \mathcal{E}(1; x, \hat{f}) & \text{if } \mathcal{I}(1; \phi, x) = \{1\} \\ \mathcal{E}(2; x, \hat{f}) & \text{otherwise} \end{cases},$$

where $\mathcal{E}(k; x, \hat{f}) := \left| \hat{f}(x_1, x_2) - \mathbb{E}[\hat{f}(X_1, X_2) \mid X_k = x_k] \right|$ for $k \in \{1, 2\}$. Therefore, the optimal order based on AUP is fully determined by $\mathcal{E}(1; x, \hat{f})$ and $\mathcal{E}(2; x, \hat{f})$, for instance, the first feature $x_1$ is more influential than the second one $x_2$ if $\mathcal{E}(1; x, \hat{f}) < \mathcal{E}(2; x, \hat{f})$. It is intuitively sensible because $\mathcal{E}(1; x, \hat{f}) < \mathcal{E}(2; x, \hat{f})$ means that the original prediction $\hat{f}(x)$ is more accurately recovered by the first feature $x_1$ than the second one $x_2$.

Using the optimal order, we demonstrate that the Shapley value does not necessarily assign a large attribution to a more influential feature. We consider the four different scenarios with two different prediction models $\hat{f}(x) \in \{1.5x_1 + x_2, 0.5x_1 + x_2\}$ and two different Gaussian distributions, $X \sim \mathcal{N}\left(\mathbf{0}_2, \Sigma_{(\rho, 2)}\right)$ for $\rho \in \{0.2, 0.6\}$. In these four scenarios, the terms $\mathcal{E}(1; x, \hat{f})$ and $\mathcal{E}(2; x, \hat{f})$ have a closed-form expression, and thus the optimal order is explicitly obtained. Moreover, due to Theorem 1, a more influential feature according to the Shapley value is explicitly obtained.

Figure 1(a) illustrates the suboptimality of the Shapley value on the four different situations. In any situation, there is a non-negligible region (described in black) where the Shapley value fails to select a more influential feature. In addition, this suboptimal area increases as the correlation gets larger, showing that the Shapley value-based explainability becomes poor when features are highly correlated.

**When there are more than two features**   When $d > 2$, it is infeasible to find the exact optimal order because there are $2^{d-1}$ possible AUPs. For this reason, we compare the Shapley value with the two commonly used marginal contribution-based methods $\Delta_1$ and $\Delta_d$, showing the Shapley value is not optimal in terms of AUP. We assume the following setting: a trained model is linear $\hat{f}(x) = \hat{\beta}_0 + x^T \hat{\beta}$ for some $(\hat{\beta}_0, \hat{\beta}) \in \mathbb{R} \times \mathbb{R}^d$ and an input vector $X = (X_1, \ldots, X_d)$ follows a Gaussian distribution $\mathcal{N}\left(\mathbf{0}_d, \Sigma_{(\rho, d)}\right)$. That is, there are $d$ features and they are equally correlated to each other with the correlation $\rho$. We set $d = 100$ and consider two different situations by varying $\rho \in \{0.2, 0.6\}$. Similar to the previous analysis, due to Theorem 1, the three attribution methods are explicitly obtained. We evaluate the prediction recovery error and the AUP on the 100 held-out test samples randomly drawn from the distribution $\mathcal{N}\left(\mathbf{0}_d, \Sigma_{(\rho, d)}\right)$. Detailed information is provided in Appendix.

Figure 1(b) illustrates the suboptimality of the Shapley value when $d = 100$ and $\rho \in \{0.2, 0.6\}$. In any situation, the prediction recovery curves for the Shapley value (described in green) have a steeper slope than $\Delta_1$ (described in red), but is not optimal as the $\Delta_d$ (described in blue) approaches to zero faster. When $\rho = 0.6$, the suboptimality becomes more severe in that the gap between $\Delta_d$ and the Shapley value gets larger.

## 4   Proposed method: WeightedSHAP

Our motivational examples in the previous section suggest that the Shapley value does not necessarily assign larger attributions for more influential features, leading to a suboptimal order of features. In fact, the last marginal contribution $\Delta_d$ outperforms other attribution methods in Figure 1(b). Although the use of $\Delta_d$ is promising, we show that focusing on one marginal contribution might lead to an unstable attribution method, suggesting a weighted mean of the marginal contributions. To be more concrete, we first examine the estimation error of the marginal contribution in the following.

**Analysis of estimation error**   In practice, the Shapley value needs to be estimated, resulting in an estimation error. Given the mathematical form of the Shapley value in (3), this estimation error arises from the estimation error of the marginal contribution. In this reason, we investigate the estimation error of the marginal contribution. We consider the same setting used in Figure 1(b) with $\rho = 0.6$. As for the estimation of the marginal contribution, we follow a standard algorithm to estimate a conditional coalition function $v^{(\mathrm{cond})}$ used in Jethani et al. [2021a] and a sampling-based algorithm to approximate $\Delta_j$. A detailed explanation for the estimation procedure and the additional result for $\rho = 0.2$ are provided in Appendix.

Figure 2 shows the relative difference between the true marginal contribution $\Delta_j$ obtained by Theorem 1 and its estimate $\hat{\Delta}_j$ as a function of the coalition size $j \in [d]$. Here, we use the relative

difference between $A$ and $B$ defined as $|A - B| / \max(|A|, |B|)$ to avoid numerical instability that can be occurred by too small marginal contribution values. It shows the $\Delta_d$, the most informative marginal contribution in Figure 1(b), has the largest relative difference from the true value. In other words, $\Delta_d$ has the largest signal to explain a model prediction, but at the same time, it is the most unstable in terms of the estimation error. This finding motivates us to consider a weighted mean of the marginal contributions that can reduce the estimation error while maintaining signals.

**Proposed method** For a weighted vector $\mathbf{w} = (w_1, \ldots, w_d)$ such that $\sum_{j=1}^{d} w_j = 1$ and $w_j \geq 0$ for all $j \in [d]$, we consider a weighted mean of the marginal contributions

$$\phi_{\mathbf{w}}(x_i) := \sum_{j=1}^{d} w_j \Delta_j(x_i). \tag{5}$$

A weighted mean $\phi_{\mathbf{w}}(x_i)$ is expected to capture the influence of features better than the Shapely value (3) by assigning a large weight to important marginal contributions. As for the game theoretic interpretation, a mathematical form of Equation (5) is known as a semivalue in cooperative game theory. It satisfies all the Shapley axioms but the efficiency axiom, which is not crucial in the attribution as we discussed in Section 2 [Dubey and Weber, 1977, Ridaoui et al., 2018]. Due to the relaxation of the efficiency axiom, a semivalue is not uniquely determined, but it is known that the semivalue is *almost* unique up to a weighted mean operation. A detailed explanation for the semivalue is provided in Appendix.

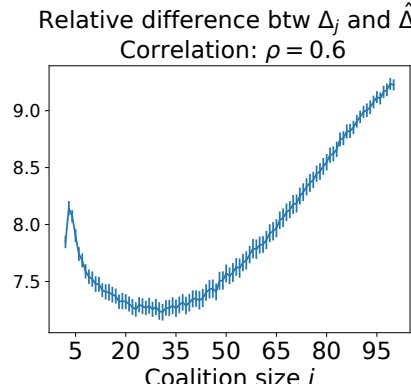

Relative difference btw $\Delta_j$ and $\hat{\Delta}_j$
Correlation: $\rho = 0.6$

Figure 2: Illustrations of the relative difference between the true marginal contribution $\Delta_j$ and its estimate $\hat{\Delta}_j$ as a function of the coalition size $j \in [d]$. We consider the same setting used in Figure 1(b). The $\Delta_d$ is shown to have the largest relative difference.

One natural and practical question that arises when using a weighted mean $\phi_{\mathbf{w}}(x_i)$ is how to select the weight vector $\mathbf{w}$. Since a weight vector to be selected is desired to have a certain good property, this question can be rephrased as to which concept of goodness should be optimized. However, it is difficult to have one universal desideratum by the intricate nature of model interpretations. There are different types of goodness and they often represent independent characteristics, as we discussed in Section 2. In other words, a good attribution essentially depends on a practitioner's downstream task. To reflect this, we propose to learn a weight vector that optimizes a user-defined utility. To be more specific, we let $\mathcal{W} \subseteq \{w \in \mathbb{R}^d : \sum_{j=1}^{d} w_j = 1, w_j \geq 0\}$ be a parametrized family of weights and $\mathcal{T}$ be a user-defined utility function that takes as input an attribution method and outputs its utility. Without loss of generality, we assume that the larger $\mathcal{T}$ is, the better it is (*e.g.*, the negative value of AUP). Given $\mathcal{T}$ and $\mathcal{W}$, we propose WeightedSHAP as follows.

$$\phi_{\mathrm{WeightedSHAP}}(\mathcal{T}, \mathcal{W}) := \phi_{\mathbf{w}^*(\mathcal{T}, \mathcal{W})}, \tag{6}$$

where $\mathbf{w}^*(\mathcal{T}, \mathcal{W}) := \mathrm{argmax}_{\mathbf{w} \in \mathcal{W}} \mathcal{T}(\phi_{\mathbf{w}})$. That is, we learn the optimal weight by optimizing a user-defined utility.

When $\mathcal{W}$ includes the uniform weight $(1/d, \ldots, 1/d) \in \mathbb{R}^d$, then by its construction, we can guarantee that WeightedSHAP is always better than or equal to the Shapley value according to the utility $\mathcal{T}$. For instance, when the negative value of AUP is used for the utility $\mathcal{T}$, AUP of WeightedSHAP is less than that of the Shapley value, *i.e.*, $\mathrm{AUP}(\phi_{\mathrm{WeightedSHAP}}) \leq \mathrm{AUP}(\phi_{\mathrm{shap}})$. Moreover, the more weight vectors are in $\mathcal{W}$, the better the quality of $\phi_{\mathrm{WeightedSHAP}}$ is guaranteed. WeightedSHAP $\phi_{\mathrm{WeightedSHAP}}$ depends on a set $\mathcal{W}$. In our experiments, we parameterize an element $\mathbf{w} \in \mathcal{W}$ by the Beta distribution inspired by mathematical properties of the semivalue in Monderer and Samet [2002, Theorem 11]. Detailed information is provided in Appendix.

**Example 1** (WeightedSHAP and the Shapley value $\phi_{\mathrm{shap}}$ on AUP). *We revisit the motivational example introduced in Section 3.2. With the negative AUP for $\mathcal{T}$ and some $\mathcal{W} \supseteq \{\Delta_d, \phi_{\mathrm{shap}}\}$, WeightedSHAP achieves significantly lower AUP than both $\Delta_d$ and $\phi_{\mathrm{shap}}$. Specifically, when $(d, \rho) = (100, 0.6)$, the AUPs of $(\Delta_d, \phi_{\mathrm{shap}}, \phi_{\mathrm{WeightedSHAP}})$ are $(1.49 \pm 0.06, 1.65 \pm 0.08, 0.77 \pm 0.03)$, respectively, where the numbers denote "mean $\pm$ standard error" based on the 100 held-out test*

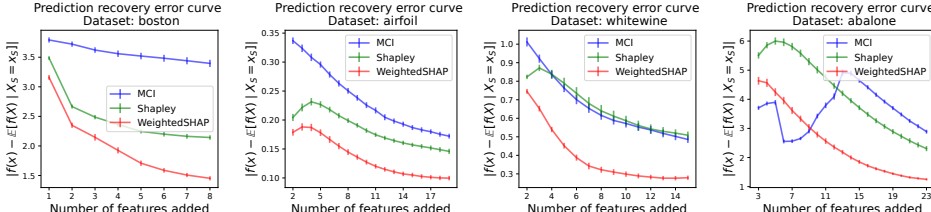

(a) Illustrations of the prediction recovery error curve on the four regression datasets. The lower, the better.

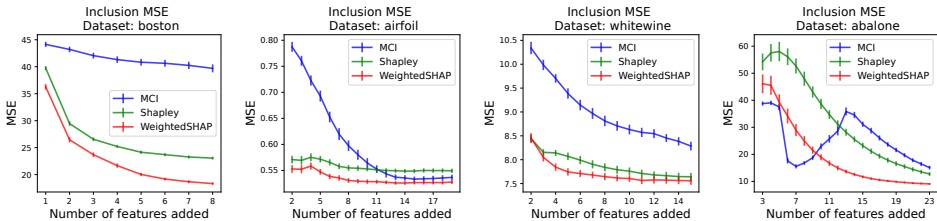

(b) Illustrations of the Inclusion MSE curve on the four regression datasets. The lower, the better.

Figure 3: **Regression tasks.** Illustrations of the prediction recovery error curve and the Inclusion MSE curve as a function of the number of features added. We add features from most influential to the least influential. We denote a 95% confidence interval based on 50 independent runs. WeightedSHAP achieves a significantly smaller MSE with fewer features than the MCI and the Shapley value.

*samples. Meanwhile, the estimation errors of ($\Delta_d$, $\phi_{\mathrm{shap}}$, $\phi_{\mathrm{WeightedSHAP}}$) are ($9.23 \pm 0.02, 6.16 \pm 0.04, 7.90 \pm 0.11$), respectively. In short, WeightedSHAP achieves a significantly lower estimation error than $\Delta_d$ while achieving the lowest AUP. Although $\phi_{\mathrm{shap}}$ achieves the lowest estimation error, its AUP is significantly greater than both $\Delta_d$ and $\phi_{\mathrm{WeightedSHAP}}$. The uniform weight in $\phi_{\mathrm{shap}}$ helps reduce the estimation error, but it loses signals too much. In contrast, WeightedSHAP well balances the signal and the estimation error, i.e., reducing the estimation error while taking more signals.*

**Implementation algorithm for WeightedSHAP** Given a finite set $\mathcal{W}$ and an easy-to-compute utility function $\mathcal{T}$, the optimal weight $\mathbf{w}^*$ can be achieved by iteratively evaluating the utility $\mathcal{T}$ for each attribution method $\phi_{\mathbf{w}}$ with $\mathbf{w} \in \mathcal{W}$. In addition, $\phi_{\mathbf{w}}$ is readily obtained as long as there are the marginal contribution estimates. Therefore, the key part of the implementation algorithm is to estimate a set of marginal contributions. The estimation of the marginal contributions consists of two parts, estimation of a conditional coalition function $v^{(\mathrm{cond})}$ and approximation of the marginal contribution $\Delta_j$. As for the first part, we train a surrogate model that takes as input a subset of input features and outputs a conditional expectation of a prediction value given the same subset [Frye et al., 2020, Jethani et al., 2021a,b]. It is known that this surrogate model unbiasedly estimates a conditional expectation of a prediction value given a subset of features under mild conditions [Frye et al., 2020, Covert et al., 2020]. Regarding the second part, a weighted mean is approximated by a sampling-based algorithm [Ghorbani and Zou, 2019, Kwon and Zou, 2021]. We provide a pseudo algorithm in Appendix. In terms of the computational cost, our algorithm is comparable to a standard the Shapley value estimation algorithm because both algorithms need to estimate the marginal contributions as a primary part [Lundberg and Lee, 2017, Frye et al., 2020]. For instance, with the classification dataset `fraud`, the marginal contribution estimation part takes 20.7 seconds per sample on average but the weight optimization part only takes 0.18 seconds, *i.e.*, the weight optimization part is only 0.86% of the total compute.

## 5 Experimental results

We demonstrate the practical efficacy of WeightedSHAP on various regression and classification datasets. We compare WeightedSHAP with the marginal contribution feature importance (MCI) proposed by Catav et al. [2021] and the Shapley value on the prediction recovery error task and the inclusion performance task. Each task assesses the goodness of an attribution order by iteratively

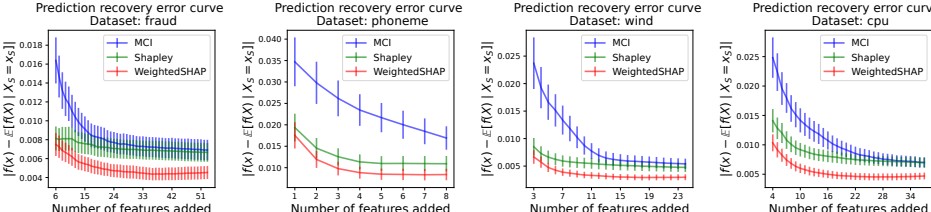

(a) Illustrations of the prediction recovery error curve on the four binary classification datasets. The lower, the better.

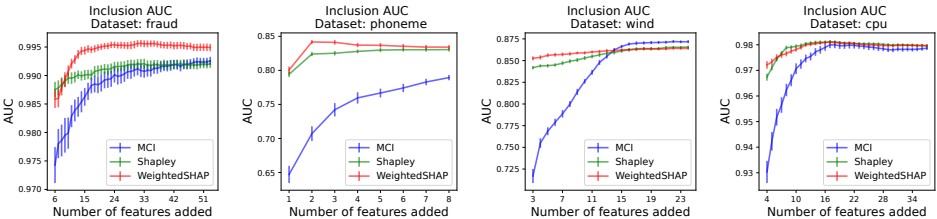

(b) Illustrations of the Inclusion AUC curve on the four binary classification datasets. The higher, the better.

Figure 4: **Classification tasks.** Illustrations of the prediction recovery error curve and the Inclusion AUC curve as a function of the number of features added. Details are provided in Figure 3. Weighted-SHAP achieves a significantly higher AUC with fewer features than the MCI and the Shapley value.

measuring how much the original model prediction or its performance is recovered with a given number of features. As for the model performance, we evaluate mean squared error (MSE) and AUC for regression and classification problems, respectively. We consider a gradient boosting model for a prediction model $\hat{f}$. As for the surrogate model in coalition function estimation $v^{(\mathrm{cond})}$, we use a multilayer perceptron model with the two hidden layers, and each layer has 128 neurons and the ELU activation function [Clevert et al., 2015]. As for the WeightedSHAP in (6), we use the negative value of the AUP for $\mathcal{T}$ and a set $\mathcal{W}$ that has 13 different weights including $\Delta_1$, $\phi_{\mathrm{shap}}$, and $\Delta_d$. All the missing details about numerical experiments are provided in Appendix, and our Python-based implementations are available at https://github.com/ykwon0407/WeightedSHAP.

Figure 3(a) compares the prediction recovery error curves for the WeightedSHAP (described in red) with the MCI (described in blue) and the Shapley value (described in green). WeightedSHAP shows always lower prediction recovery errors than the MCI and the Shapley value. Given that WeightedSHAP minimizes the AUP, which is the sum of prediction recovery error $|\hat{f}(x) - \mathbb{E}[\hat{f}(X) \mid X_S = x_S]|$, WeightedSHAP does not necessarily have a smaller prediction recovery error for every number of features added. As for the MSE, Figure 3(b) shows that WeightedSHAP has a significantly smaller MSE than baseline methods with fewer features. In particular, on the airfoil dataset, WeightedSHAP achieves 0.53 MSE with 10 features, but the Shapley value never achieves this value because of the suboptimality of the attribution order.

We also evaluate the prediction recovery error and AUC for the four classification datasets. Similar to the regression cases, Figures 4(a) and 4(b) show that WeightedSHAP effectively assigns larger values for more influential features and recovers the original prediction $\hat{f}(x)$ significantly faster than the MCI and the Shapley value. Specifically, on the fraud dataset, WeightedSHAP achieves 0.995 AUC with 14 features, but the Shapley value always has the lower AUC value. Our findings are consistently observed with a different model for $\hat{f}$ or other datasets. Additional experimental results with different evaluation metrics and a qualitative assessment of WeightedSHAP are provided in Appendix.

## 5.1 Illustrative examples from MNIST

We present a qualitative assessment of WeightedSHAP and examine how its top influential features differ from those from SHAP using the MNIST dataset. We train a convolutional neural network model using the same setting suggested in Jethani et al. [2021a]. It achieves 98.6 % accuracy

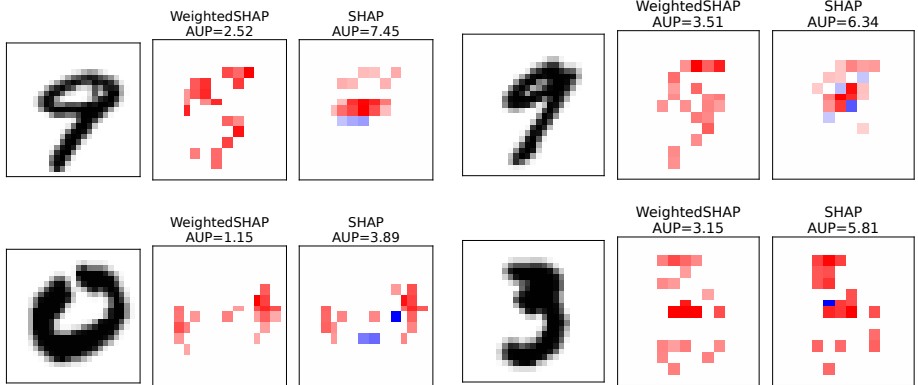

Figure 5: Illustrative examples of WeightedSHAP and SHAP attributions on MNIST images. We present the top 10% of influential features. Red (*resp.* blue) color indicates the corresponding feature positively (*resp.* negatively) affects the model prediction. (Top) WeightedSHAP clearly captures the last stroke of nine while SHAP fails to capture it. (Bottom) While SHAP has noisy negative feature attributions described by blue pixels, WeightedSHAP provides noiseless and intuitive explanations.

on the test dataset. We select illustrative images with a significant difference in AUPs between WeightedSHAP and SHAP.

Figure 5 compares the top 10% influential attributions for WeightedSHAP and the Shapley value. On the top images, while SHAP fails to capture the last stroke of digit nine, which is a crucially important stroke to differentiate from the digit zero, WeightedSHAP clearly captures the strokes. On the bottom images, SHAP produces unintuitive negative attributions, providing noisy explanations. In contrast, WeightedSHAP provides noiseless and intuitive explanations.

## 6   Conclusion

In this paper, we provide an analysis of the widely used SHAP attribution method. We discover that even in simple natural settings, SHAP can incorrectly identify important features. Mathematically, a key limitation of SHAP is in that it assigns uniform weights to all marginal contributions. We propose WeightedSHAP which generalizes the Shapley value by relaxing the efficiency axiom. WeightedSHAP learns to pay more attention to the marginal contributions that have more signal on a prediction, assigning larger attributions for more influential features. There are several limitations of WeightedSHAP that motivate interesting future works. Here we use the AUP metric to optimize the weights because AUP is commonly used in practice. However, there is no agreed-upon metric for evaluating feature attribution methods. Different users may care about different notions of attribution. Developing variants of marginal contribution weighting optimized for different applications is an important direction of future research. We believe that the core contribution of this paper—that the uniform weighting used by SHAP can be suboptimal—still provides useful insights for these investigations. Here we focus our experiments on directly comparing WeightedSHAP with SHAP because our goal is to characterize the limitations of SHAP. There is a large body of works comparing SHAP with other attribution methods that are complementary to our work [Yeh et al., 2019, Jethani et al., 2021a].

## Acknowledgment

The authors would like to thank all anonymous reviewers for their constructive comments. We also would like to thank Ruishan Liu for the helpful discussion.

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
