# OpenReview forum: "WeightedSHAP: analyzing and improving Shapley based feature attributions"
_NeurIPS.cc/2022/Conference — NeurIPS 2022 Accept_

### Official Review · Reviewer_hhZc · 2022-06-28

**Rating:** 6
**Confidence:** 4
**Soundness:** 3 good
**Presentation:** 3 good
**Contribution:** 3 good

**Summary:**

The paper illustrates that regular SHAP does yield a good AUP. They provide examples of why this is the case and provide arguments that regular SHAP is suboptimal. They propose WegihtedSHAP as a solution to improve the AUP, which parametrizes the marginal contributions with a weight that is optimized on the user-specified utility function $\mathcal{T}$. Experiments the demonstrate the advantage of WeightedSHAP.

**Questions:**

1. It's not clear what $\mathcal{T}$ exactly means, can you give an example?
2. I've understood that suboptimal refers to that AUP "converges" slower when you add the top k important features. Would it be possible to characterize this in terms of a convergence rate?

------

Post-rebuttal comments:

I'd like to thank the authors for their response and clarification. Happy to keep my score at 6, I think the image explanations are a nice addition.



**Limitations:**

The paper considers tabular data for regression and classification. It would be interesting to understand if this could be extended to consider image data as well.

**Strengths And Weaknesses:**

Positive points:
1. Clearly defined problem - It's well-formulated through empirical evidence
2. Solution is quite simple
3. Experiments convincing

Negative points:
1. Writing could be clearer

---

> ### Author Response · Authors · 2022-08-02
> **Response to Reviewer hhZc**
>
> We appreciate the reviewer for the positive comments!
>
> **Regarding $\mathcal{T}$:** A function $\mathcal{T}$ takes as input a $d$-dimensional vector of feature attributions and outputs a univariate user-defined utility that can represent the goodness of the attributions. The higher $\mathcal{T}(\phi)$ is, the better $\phi$ is. In our experiments, the negative value of $\mathrm{AUP}(\phi)$ is used for $\mathcal{T}(\phi)$, following the literature.
>
> **The top k features and the convergence rate:** Thank you for this question! In the attribution problem, it is desirable to know what the $k$ features recover the original prediction most. However, as we showed in motivational examples in Section 3.2, it is often infeasible to specify the top $k$ important features even in simple settings. This makes analysis of ‘convergence rate’ difficult, but we believe it is an interesting and impactful future topic.
>
> **Regarding image data:** We have added clarifications throughout the revised text. WeightedSHAP is a general method that can be used on any data where regular Shapley is used, including image data. Following reviewer SvbQ’s suggestion, we have added qualitative examples demonstrating applications of WeightedSHAP to images in Figure 9 (Appendix E).

---

### Official Review · Reviewer_SvbQ · 2022-07-09

**Rating:** 6
**Confidence:** 4
**Soundness:** 2 fair
**Presentation:** 3 good
**Contribution:** 2 fair

**Summary:**

The paper investigates a limitation in the feature attribution method SHAP. To ameliorate this limitation, the authors propose a method WeightedSHAP, which generalizes SHAP to use a weighted average over coalition sizes. The optimal weighting is chosen based on a given utility function and set of possible weights. Experiment results show that the attributions generated WeightedSHAP have better performance than Shapley values on the AUP metric.

**Questions:**

* The experiment in Figure 2 shows the relative difference between $\hat \Delta_j$ and $\Delta_j$, but how is this difference reduced by using the weighted mean? Is there a corresponding figure plotting the relative difference after applying the weighted mean?

**Limitations:**

Limitations have been adequately discussed in the main text.

**Strengths And Weaknesses:**


**Strengths:**
* The investigated problem is important; SHAP is a popular method for approximating feature attributions. The proposed method shows improvement over the baseline method on a key metric, and has a nice game-theoretic interpretation.
* The paper is well-written and well-motivated, with clear and concise notation.

**Weaknesses:**
* Limited comparisons. The only comparison is to the unweighted Shapley Value on AUP. The paper could be improved by including other metrics and/or value functions. For example, it would be interesting to see if improved performance on AUP generalizes to performance on other metrics (e.g. ROAR [1]). There are also other methods that similarly aim to improve the Shapley value's limitations w.r.t. feature correlation. For example, [2] proposes a modified value function. [3] proposes a feature attribution ranking that incorporates feature interactions. Comparisons to such methods would improve the paper.
* No qualitative analysis. The proposed method shows improvement on AUP, however it would be beneficial to see how the actual feature importance ordering changes. Ideally there would be some user study, but even qualitative examples would improve the paper
* No time comparison or time complexity analysis. The main text states that the "[computation cost] is comparable to the Shapley value estimation algorithm", however the WeightedSHAP algorithm must add some nonzero computation expense which likely increases as size of $\mathcal{W}$ and / or number of features increases. The experiments also use relatively small datasets and |\mathcal{W}| seems relatively small.

**Miscellaneous Minor Issues:**
* Typo in Line 132: "efficiency axiom does is not required"
* Line 279: "WeightedSHAP is always better than the Shapley value". Minor nitpick, however this statement could be misleading, given that it's possible for WeightedSHAP to equal performance of Shapley value (as stated in the next sentence).

**Summary:**
The paper is well-motivated, and the proposed method shows improvement over the unweighted Shapley value on the AUP metric. I have concerns with the limited experiments (see above); including additional qualitative comparisons and comparisons with other metrics would improve on this point.


[1] Hooker et al, A Benchmark for Interpretability Methods in Deep Neural Networks
[2] Catav et al, Marginal Contribution Feature Importance - an Axiomatic Approach for   Explaining Data
[3] Masoomi et al, Explanations of Black-Box Models based on Directional Feature Interactions

---

> ### Author Response · Authors · 2022-08-02
> **Response to Reviewer SvbQ**
>
> We appreciate the reviewer for the valuable comments and the references.
>
> **Regarding additional comparisons:** We greatly thank the reviewer for this helpful suggestion! We have incorporated new experimental results with the methods the reviewer suggested (Catav et al. (2021); Masoomi et al. (2021)). Please see the general response.
>
> [References]
> - Catav, A., Fu, B., Zoabi, Y., Meilik, A. L. W., Shomron, N., Ernst, J., ... & Gilad-Bachrach, R. (2021, July). Marginal contribution feature importance-an axiomatic approach for explaining data. In International Conference on Machine Learning (pp. 1324-1335). PMLR.
> - Masoomi, A., Hill, D., Xu, Z., Hersh, C. P., Silverman, E. K., Castaldi, P. J., ... & Dy, J. (2021, September). Explanations of Black-Box Models based on Directional Feature Interactions. In International Conference on Learning Representations.
>
> **Regarding qualitative analysis:** Thank you for pointing out this issue. We have included illustrative examples using the MNIST dataset. We compare the top $10\%$ influential features of WeightedSHAP with SHAP, demonstrating that WeightedSHAP is promising to generate attributions that are more intuitive and noiseless. The illustrative examples are provided in Figure 9 (Appendix E).
>
> **Adding time computational analysis:** Although it is true that our algorithm needs nonzero computational costs due to the weight optimization, we would like to emphasize that the main computational cost is from the marginal contribution estimation part, not from the weight optimization. For instance, with the classification dataset $\mathsf{fraud}$, the marginal contribution estimation part takes $20.70$ seconds per sample on average but the weight optimization part only takes $0.18$ seconds. Hence, the weight optimization part is only $0.86$ % of the total costs. We have added this note at the end of Section 4 (page 8 of the revision).
>
> **Editorial issues:** Thank you for pointing out these issues. We have incorporated them in the revision.
>
> **WeightedSHAP’s behavior in Figure 2:** Please see the general response.
>
> We again thank Reviewer SvbQ for constructive and helpful comments. Your questions have greatly improved our work. We hope you would consider improving your score in light of our detailed response.

---

> > ### Comment · Reviewer_SvbQ · 2022-08-06
> > **Response**
> >
> > Thank you for your responses. I appreciate the changes you've made to the paper which have addressed my main concerns, particularly those regarding additional comparisons and metrics. The illustrative examples are also very nice. I will therefore increase my score to 6.

---

> > > ### Author Response · Authors · 2022-08-06
> > > **Thank you**
> > >
> > > We are glad that we addressed your concerns. We also thank you for re-evaluating our work!

---

> ### Author Response · Authors · 2022-08-05
> **We would like to hear back from reviewer SvbQ**
>
> Dear reviewer SvbQ,
>
> We thank the reviewer again for the constructive review. We would like to follow up to see if our response addresses your concerns or if you have any further questions. We would really appreciate the opportunity to discuss this further if our response has not already addressed your concerns. Thank you again!

---

### Official Review · Reviewer_AA8w · 2022-07-12

**Rating:** 6
**Confidence:** 4
**Soundness:** 3 good
**Presentation:** 3 good
**Contribution:** 3 good

**Summary:**

The paper considers the task of feature attribution (specifically, from the perspective of correctly capturing the *order* of attributions). It demonstrates that a common feature attribution method based on the Shapley value (SHAP) is sub-optimal in the sense that there exist instances in which it does not correctly identify the order of important features. It proposes a modification of SHAP that relaxes the efficiency axiom, where the marginal contributions can be weighted arbitrarily, and the optimal weighing is learned from data.



**Questions:**

1. The current paper seems to suggest that ideally (if estimation weren't an issue), we would assign higher importance to $\Delta_d$ than the importance assigned by the Shapley value. If I understand correctly, this seems to contradict part of the motivation for removing the efficiency axiom in Kwon & Zou (they show $\Delta_j$ becomes less informative as $j$ approaches $d$, mostly because the marginal contribution of a single element to a large training data diminishes). Can you clarify this point?

2. Not only does the assumption that the set $W$ is finite seems limiting, but I was surprised to see that the cardinality of $W$ in the experiments was very small (13!). Several questions: Was $W$ chosen to be so small for computational reasons, i.e. because the algorithm is basically a for loop over the elements in $W$? How did you select the set $W$? Are the results sensitive to this choice? (the paper points to the appendix for details but I could not locate them).

3. Can you demonstrate how weightedSHAP behaves on the examples from Section 3? (see above)

**Limitations:**

Yes

**Strengths And Weaknesses:**

Strengths: The paper is written clearly and related work is discussed properly. Understanding the limitations of current feature attribution methods is an important area of research.

Weaknesses:

1. One general weakness of this line of work is that by questioning whether the assumptions made by the Shapley value are relevant to the objective of correctly identifying the order of feature attributions ("given that Shapley axioms are often not readily applicable to ML problems"), the choice to stay within the general framework of Shapley seems more questionable. I understand that the efficiency axiom doesn't seem important for correctly identifying the order of important features - is the linearity axiom any more relevant than that? If one could write a similar paper proposing to relax the linearity axiom, then isn't just the takeaway here that the entire approach of using the Shapley value is not optimal? To be clear, this isn't an argument specifically against the current paper, but I do think it's important to keep in mind when considering the contributions & impact of the paper.

2. I think the connection between the observations regarding sub-optimality of SHAP (Section 3) and the proposed method (Section 4) could be articulated better. E.g. the main takeaway from the first negative example in Section 3 is that SHAP fails when the features are highly correlated. How does allowing the weights of the marginal contributions to be non-uniform address this issue?  I'm not sure I understand how the explanation regarding the issue of estimation error in the beginning of Section 4 is technically related to the negative results of Section 3. Ideally, it would be good to revisit the motivational examples from Section 3 and demonstrate how weightedSHAP improves on them (e.g. reduces the region of failure).

---

> ### Author Response · Authors · 2022-08-02
> **Response to Reviewer AA8w**
>
> We greatly appreciate the reviewer for the detailed comments and suggestions.
>
> **Assumptions made by Shapley value:** Thank you for the thought-provoking comment. We agree that the assumptions made by the Shapley value are often questionable in attribution problems as they are originally founded in cooperative game theory literature. This actually motivated our paper as we highlight the limitations of the standard Shapley attribution method. The Shapley value is one of the most commonly used attribution methods in practice. Therefore, it is still very valuable to understand its limitations and to propose WeightedSHAP as an easy-to-implement modification that performs better than the standard Shapley.
>
> **Regarding the most important marginal contributions:** You are correct. Based on our empirical experiments, marginal contributions based on a large coalition size capture influential features better than a small one. This is opposite to the case shown in data valuation problems, where marginal contributions based on a small coalition size are more attractive choices than a large one. In comparison between attribution problems and data valuation problems, we find that a statistical theory in data valuation problems is not readily applied to attribution problems: the U-statistics theory used in Theorem 1 of Kwon and Zou (2022) is not applied in attribution problems, as the underlying data distribution changes whenever a subset of features changes.
>
> [Reference]
>
> - Kwon, Y., & Zou, J. (2021). Beta shapley: a unified and noise-reduced data valuation framework for machine learning. arXiv preprint arXiv:2110.14049.
>
> **Regarding the choice of $\mathcal{W}$:** The main reason we used such a set $\mathcal{W}$ is *NOT* because of the computational costs. Although it is true that our algorithm needs to repeat over the elements in the set $\mathcal{W}$, we would like to emphasize that the main computational cost is from the marginal contribution estimation part, not from the weight optimization. For instance, with the classification dataset $\mathsf{fraud}$, the marginal contribution estimation part takes $20.70$ seconds per sample on average, but the weight optimization part only takes $0.18$ seconds. Hence, the weight optimization part is only $0.86$ % of the total compute. Increasing the size of $\mathcal{W}$ does not significantly increase compute. As for the small cardinality, we find that our selection $\mathcal{W}$, a set of 13 feature attribution methods, actually yields wide ranges on AUPs, and accordingly, it is enough to show the suboptimality of the Shapley value. As for the sensitivity, it depends on the selection of $\mathcal{W}$, but at the same time, we believe the performance of WeightedSHAP would increase as one uses a larger set for $\mathcal{W}$. Finally, our experiments show that even a modest $\mathcal{W}$ is already sufficient to substantially improve performance compared to Shapley across commonly used metrics.
>
> **WeightedSHAP’s behavior in Figure 2:** Thank you for this question! Please see the general response.

---

### Official Review · Reviewer_E2Dv · 2022-07-16

**Rating:** 4
**Confidence:** 4
**Soundness:** 3 good
**Presentation:** 3 good
**Contribution:** 1 poor

**Summary:**

This paper found that the conditional expectation-based Shapley value might fail in some cases, and they thought the problem was caused by the fact that the Shapley value uses the same weight for all marginal contributions. They proposed WeightedSHAP where they assigned a learnable weight to the marginal contributions and make the WeightedSHAP equal to the weighted average of the marginal contributions.

**Questions:**

N/A

**Ethics Review Area:**

["I don’t know"]

**Strengths And Weaknesses:**

[Strength]
1. This paper showed a significant problem of the conditional expectation-based Shapley value and provided detailed theoretical discussion about it.

[Weakness]
1. This paper only discussed about the conditional expectation-based Shapley (CES) value, while baseline value-based Shapley (BShap) value was not discussed. [cite 1] has already showed that the CES didn't satisfy the dummy and linearity axioms, while BShap could satisfy all four axioms. Therefore, it is not surprising that the CES might fail in some special cases.

[cite 1] Mukund Sundararajan and Amir Najmi. The many shapley values for model explanation. In International conference on machine learning, pages 9269–9278. PMLR, 2020

2. The WeightedSHAP lacks of theorectical soundness. In addition to efficiency axiom, due to the nature of CES, the weightedSHAP also does not satisfy the dummy and linearity axioms, thus this method does not satisfy three of the four axioms of the Shapley value. And, the weight vector of weightedSHAP are learned from some utility function which might cause bias towards the model explanation and influence the objectivity of the explanation. Therefore, the WeightedSHAP lacks of theorectical soundness.
3. The experimental results in this paper may not be objective. The weight vector of the WeightedSHAP is optimized w.r.t the utility function AUP, while some similar metric AUC or MSE curve are used to compare the Shapley and WeightedSHAP.
4. The experiment was not sufficient. Other than the method proposed in [cite 2], some other Shapley-value based explaination methods (like DeepSHAP [cite 3], etc.) should be also compared with WeightedSHAP.

[cite 2] Christopher Frye, Damien de Mijolla, Tom Begley, Laurence Cowton, Megan Stanley, and Ilya Feige. Shapley explainability on the data manifold. arXiv preprint arXiv:2006.01272, 2020.

[cite 3] Scott M Lundberg and Su-In Lee. A unified approach to interpreting model predictions. In Advances in neural information processing systems, pages 4765–4774, 2017.

---

> ### Author Response · Authors · 2022-08-02
> **Response to Reviewer E2Dv (1/2)**
>
> We thank the reviewer for the thoughtful comments and suggestions.
>
> **The Shapley axioms and conditional expectation-based Shapley (CES):** The reviewer claimed that the CES does not satisfy the Dummy and Linearity axioms, but this is not true. Given that the only difference between CES and baseline value-based Shapley (BShap) is the coalition function, *both CES and BShap satisfy the four standard Shapley axioms*: Linearity, Dummy, Symmetry, and Efficiency (Also see our response on the **Theoretical soundness of WeightedSHAP**).
>
> **Theoretical soundness of WeightedSHAP:** We would like to clarify a few misunderstandings the reviewer had about the Dummy and Linearity axioms, and then we explain that WeightedSHAP satisfies the Dummy and Linearity axioms.
>
> + **Regarding the Dummy and Linearity axioms:** As the reviewer mentioned, Remark 4.13 of Sundarajan and Najmi (2020) showed that the CES fails to satisfy Dummy and Linearity axioms. However, we would like to highlight that their Dummy and Linearity axioms are **DIFFERENT** from the one used in our paper. The axioms we used are based on Shapley (1953), which are often considered more *standard* in cooperative game theory literature (Banzhaf III, 1964; Weber, 1988) and in machine learning literature (Covert et al. 2021; Rozemberczki et al. 2022). To be more specific, the axioms in Sundarajan and Najmi (2020) are described by the property of *a function $f$*, but the axioms we used are described by the property of *a set function $v$*, which depends on both the function $f$ and the distribution $D$. This difference is actually discussed in the second paragraph of Section 3.3 Sundarajan and Najmi (2020), saying “the various axioms still hold for $v$ based on the conditional expectation.” In other words, although Sundarajan and Najmi (2020) were able to claim that CES fails to satisfy their function-based Dummy and Linearity axioms, this is not true for the **standard** Shapley axioms.
>   + With the **standard** Shapley axioms, the CES indeed satisfies the four Shapley axioms including Dummy and Linearity axioms because its functional form is the Shapley value. We have presented the definitions of the axioms in Appendix B of the previously submitted Supplementary Material.
>
> + **Shapley axioms and WeightedSHAP:** As we presented in Proposition 3 in Appendix B of the previously submitted Supplementary Material, it is known that every semivalue satisfies the *standard* Linearity, Null-player (also known as Dummy), and Symmetry axioms. Hence, as a type of semivalue, the WeightedSHAP satisfies all these three axioms, having rigorous theoretical foundations.
>
> [References]
>
> - Sundararajan, M., & Najmi, A. (2020, November). The many Shapley values for model explanation. In International conference on machine learning (pp. 9269-9278). PMLR.
> - Shapley, L. (1953). A Value for n-person Games. Contributions to the Theory of Games II, Kuhn, H., Tucker, A.
> - Banzhaf III, J. F. (1964). Weighted voting doesn't work: A mathematical analysis. Rutgers L. Rev., 19, 317.
> - Weber, R. J. (1988). Probabilistic values for games. The Shapley Value. Essays in Honor of Lloyd S. Shapley, 101-119.
> - Covert, I., Lundberg, S. M., & Lee, S. I. (2021). Explaining by Removing: A Unified Framework for Model Explanation. J. Mach. Learn. Res., 22, 209-1.
> - Rozemberczki, B., Watson, L., Bayer, P., Yang, H. T., Kiss, O., Nilsson, S., & Sarkar, R. (2022). The Shapley Value in Machine Learning. arXiv preprint arXiv:2202.05594.

---

> ### Author Response · Authors · 2022-08-02
> **Response to Reviewer E2Dv (2/2)**
>
> **The objectivity of experimental results:** We thank you for the comment. There is not an agreed-upon objective metric for ML interpretability. We have used standard evaluation metrics used in many recent publications. Given that the concept of *influential* features can be dependent on downstream tasks, we believe the attribution should be optimized to downstream objectives. For instance, there is no single attribution method that works universally well simultaneously on both the Inclusion AUC and Exclusion AUC tasks, and the optimal attribution depends on the evaluation metric. In this sense, our WeightedSHAP is very flexible in optimizing downstream objectives and finds the most suitable attributions. In contrast, the Shapley value is fixed to every downstream task.
>
> **Regarding insufficient experimental results:** Thank you for pointing out this issue! We have included a new attribution method, MCI, in the revision and have incorporated additional numerical experiments in Appendix D.4. WeightedSHAP substantially outperforms MCI and the standard Shapley method across commonly used metrics. Please see the general response.
>
> We again thank Reviewer E2Dv for your time and valuable comments. We hope our responses on the Shapley axioms resolve your concerns about the theoretical soundness of WeightedSHAP. We hope you would consider improving your score in light of our detailed response. Please do not hesitate to let us know if you have any questions. We will be very happy to follow up.

---

> ### Author Response · Authors · 2022-08-05
> **We would like to hear back from reviewer E2Dv**
>
> Dear reviewer E2Dv,
>
> We would like to follow up to see if our response addresses your concerns or if you have any further questions. We would really appreciate the opportunity to discuss this further if our response has not already addressed your concerns. Thank you again!

---

### Author Response · Authors · 2022-08-02
**Response to all reviewers**

We thank all the reviewers for their time and constructive feedback on our work. In the revised paper that we have uploaded, we have carefully incorporated the reviewers’ helpful suggestions. Also, we respond to each of the reviewer’s comments and the main additions are summarized as follows.

**WeightedSHAP’s behavior in Figure 2:** Reviewers AA8w and SvbQ asked about the WeightedSHAP’s behavior on the motivational example used in Figure 2. In the revision, we have further examined the analysis in Figure 2 with the same hyperparameter pair $(\mathcal{T}, \mathcal{W})$ used in Section 5. We observe that WeightedSHAP achieves a lower AUP than $\Delta_d$ and the Shapley value while avoiding large estimation errors (lower AUP corresponds to better attributions). We have incorporated this note in Example 1 (page 7 of the revision).

**Regarding additional numerical experiments:** Reviewers E2Dv and SvbQ recommended additional numerical experiments. Based on this, we have added a new attribution method, the marginal contribution feature importance (MCI) proposed by Catav et al. 2021, to every experiment we conducted in Section 5. Our observations show that WeightedSHAP achieves a significantly better performance than the MCI and the Shapley value on inclusion performance tasks.

Furthermore, we have incorporated two new evaluation metrics: the Exclusion performance used in Jethani et al. (2022) and the Inclusion performance with masked features used in Masoomi et al. (2021). Our experiments show that WeightedSHAP achieves a significantly better performance than baseline methods for various downstream tasks and that the standard Shapley value is suboptimal, verifying our main findings. Lastly, we have included illustrative examples using the MNIST dataset, showing that WeightedSHAP is more effective to capture noiseless and intuitive explanations than the Shapley value. We have added all new experimental results in Figures 8 and 9 (Appendix D.4 and E).


[References]

- Catav, A., Fu, B., Zoabi, Y., Meilik, A. L. W., Shomron, N., Ernst, J., ... & Gilad-Bachrach, R. (2021, July). Marginal contribution feature importance-an axiomatic approach for explaining data. In International Conference on Machine Learning (pp. 1324-1335). PMLR.
- Jethani, N., Sudarshan, M., Covert, I., Lee, S. I., & Ranganath, R. (2021). FastSHAP: Real-Time Shapley Value Estimation. arXiv preprint arXiv:2107.07436.
- Masoomi, A., Hill, D., Xu, Z., Hersh, C. P., Silverman, E. K., Castaldi, P. J., ... & Dy, J. (2021, September). Explanations of Black-Box Models based on Directional Feature Interactions. In International Conference on Learning Representations.

---

### Meta-Review · Area_Chair_JKFx · 2022-08-24

**Recommendation:** Accept
**Confidence:** Certain

**Metareview:**

Strengths:
* paper points out and addresses significant issue in conventional Shapley-based approaches
* theoretical analysis having game-theoretic interpretation
* convincing empirical results (after revision)
* clear writing, good survey of related work

Weaknesses:
* limited empirical comparisons (addressed in revision)
* lacks qualitative analysis (addressed in revision)
* raises general questions regarding suitability of Shapley approaches, and the role of assumptions within


Summary:
This paper presents a nice generalization of Shapley-based feature attribution that, by introducing weights, mitigates a certain drawback of the standard approach, and provides more flexibility. Most reviewers view the paper’s contributions favorably; theoretical results and interpretations appear to be sound, and the empirical results complement them well (after additions made by the authors in the revised version and in response to reviewer feedback). One reviewer was worried about the soundness of a certain aspect of the proposed approach, but the authors’ response was helpful in clarifying this issue. Another reviewer pointed out that, more generally, arguing for relaxing assumptions (e.g., via weighting) may actually suggest that the overall approach is limited; the authors are encouraged to add a discussion in the paper that addresses this concern.


**Award:**

No

---

### Decision · Program_Chairs · 2022-09-14

Accept